



# Enhancing High-Resolution Forest Stand Mean Height Mapping in China through an Individual Tree-Based Approach with Close-Range LiDAR Data

Yuling Chen[1],[★], Haitao Yang[1],[★], Zekun Yang[1], Qiuli Yang[2,3], Weiyan Liu[4], Guoran Huang[5], Yu Ren[1],
Kai Cheng[1], Tianyu Xiang[6], Mengxi Chen[1], Danyang Lin[4], Zhiyong Qi[1], Jiachen Xu[1], Yixuan Zhang[1],
Guangcai Xu[7], Qinghua Guo[1,8*]

[1]Institute of Remote Sensing and Geographic Information System, School of Earth and Space Sciences, Peking University, Beijing 100871, China
[2]College of Geography and Remote Sensing Science, Xinjiang University, Urumqi 800017, China
[3]Xinjiang Key Laboratory of Oasis Ecology, Xinjiang University, Urumqi 830017, China
[4]State Forestry and Grassland Administration Key Laboratory of Forest Resources & Environmental Management, Beijing Forestry University, Beijing 100083, China
[5]College of Forestry, Southwest Forestry University, Kunming 650224, China
[6]College of Earth Sciences, Chengdu University of Technology, Chengdu 610059, China
[7]Beijing GreenValleyTechnology Co., Ltd, HaidianDistrict, Beijing 100091, China
[8]Institute of Ecology, College of Urban and Environmental Sciences, Peking University, Beijing 100871, China

★These authors contributed equally to this work.

*Correspondence to*: Qinghua Guo (guo.qinghua@pku.edu.cn)

**Abstract.** Forest stand mean height is a critical indicator in forestry, playing a pivotal role in various aspects such as forest inventory estimation, sustainable forest management practices, climate change mitigation strategies, monitoring of forest structure changes, and wildlife habitat assessment. However, there is currently a lack of large-scale, spatially continuous forest stand mean height maps. This is primarily due to the requirement of accurate measurement of individual tree height in each
forest plot, a task that cannot be effectively achieved by existing globally covered, discrete footprint-based satellite platforms. To address this gap, this study was conducted using over 1117 km$^2$ of close-range Light Detection and Ranging (LiDAR) data, which enables the measurement of individual tree height in forest plots with high precision. Besides, this study incorporated spatially continuous climatic, edaphic, topographic, vegetative, and Synthetic Aperture Radar data as explanatory variables to map the tree-based arithmetic mean height ($h_a$) and weighted mean height ($h_w$) at 30 m resolution across China. Due to
limitations in obtaining basal area of individual tree within plots using UAV LiDAR data, this study calculated weighted mean height through weighting an individual tree height by the square of its height. In addition, to overcome the potential influence of different vegetation divisions at large spatial scale, we also developed a machine learning-based mixed-effects model to map forest stand mean height across China. The results showed that the average $h_a$ and $h_w$ across China were 11.3 m and 13.3 m with standard deviations of 2.9 m and 3.3 m, respectively. The accuracy of mapped products was validated utilizing LiDAR
and field measurement data. The correlation coefficient ($r$) for $h_a$ and $h_w$ ranged from 0.603 to 0.906 and 0.634 to 0.889,





while RMSE ranged from 2.6 to 4.1 m and 2.9 to 4.3 m, respectively. Comparing with existing forest canopy height maps derived using the area-based approach, it was found that our products of $h_a$ and $h_w$ performed better and aligned more closely with the natural definition of tree height. The methods and maps presented in this study provide a solid foundation for estimating carbon storage, monitoring changes in forest structure, managing forest inventory, and assessing wildlife habitat

availability. The dataset constructed for this study is publicly available at https://doi.org/10.5281/zenodo.12697784 (Chen et al., 2024).

## 1 Introduction

Tree height is a pivotal indicator in forestry (Wang et al. 2019a), with paramount importance for forest inventory, wildlife habitat assessment and climate change mitigation strategies (Vaglio Laurin et al. 2019; Wang et al. 2019b; Zemp et al. 2023).

Forest stand height denotes the mean height of trees within a stand/plot, including arithmetic mean height and mean height weighted in proportion to their basal area (weighted mean height or Lorey's mean height) (Laar and Akça 2007; Masaka et al. 2013). It serves as a key factor in assessing forest growth (Ma et al. 2023; McGregor et al. 2021), calculating forest volume (Xu et al. 2019) and carbon storage (Yao et al. 2018), as well as guiding sustainable forest management practices (Xu et al. 2023). Nevertheless, traditional tree height measurements derived from field surveys are typically time-consuming and

resource-intensive (Jurjević et al. 2020; Liu et al. 2022), making it impractical to generate comprehensive wall-to-wall forest stand mean height data products across extensive spatial scales (Su et al. 2017). Although passive remote sensing techniques offer a potential solution for estimating forest stand height indirectly (Donoghue and Watt 2006; Hall et al. 2006; Lu et al. 2004; Zhang et al. 2014). They are constrained by penetration ability and saturation effects, resulting in inherent uncertainty issues (Liu et al. 2022; Su et al. 2015). Mapping high-resolution forest stand mean height at a large scale through individual

tree-based measurements remains a challenging but crucial objective in forest management and ecosystem monitoring.

Light detection and ranging (LiDAR), utilizing focused wavelength laser pulses, is an active remote sensing technique renowned for its robust penetration ability to directly characterize three-dimensional structures of forest (Guo et al. 2021; Liang et al. 2022; Liu et al. 2022; Ma et al. 2023; Su et al. 2017). The development of multi-platforms LiDAR scanning has greatly enhanced the precision of tree height measurement from the individual trees to regional scales (Guo et al. 2021; Jurjević

et al. 2020; Liu et al. 2022; Wang et al. 2019a). Additionally, to address challenges in accuracy and cost of LiDAR data, the innovations in close-range LiDAR, particularly through the use of unmanned aerial vehicles (UAVs) and terrestrial laser scanning, have enhanced the flexibility, accessibility and cost-effectiveness of LiDAR data in forestry applications (Guo et al. 2021; Hu et al. 2021; Yin et al. 2024). Consequently, the advancement of close-range LiDAR has laid a robust foundation of data and technology, facilitating the high-resolution wall-to-wall tree height mapping at large spatial scales.

Two main approaches are utilized for tree height measurement with LiDAR data: area-based approach (Bouvier et al. 2015; Liu et al. 2022); and tree-based approach (Su et al. 2017; Swayze et al. 2021; Yin et al. 2024). These approaches differ in their definition of "height" and the method used for calculation. The area-based approach, also known as the grid-based approach,



simplifies the process of obtaining tree height. It involves generating a canopy height model (CHM) to calculate tree height based on the statistical relationships between plot-level LiDAR metrics. The height metrics from obtained from this approach

is forest canopy height, which include not only the actual tree height, but also the height of other branches (Bouvier et al. 2015). Additionally, changes of in height within non-tree or non-vegetation pixels may also be included, leading to further deviation from the definition of tree height in forestry (Yin et al. 2024). Tree height calculated from the tree-based approach closely aligns with to the natural definition of tree height, which the height being evaluated is the height from individual tree, ranging from the treetop to the ground. This approach requires first detecting individual trees in a sample plot, the measuring

the height of each tree in this plot, and finally calculating forest stand height (Laar and Akça 2007; Masaka et al. 2013). Data from close-range LiDAR, with its advanced algorithms for individual tree segmentation, is widely used in the tree-based approach (Li et al. 2012; Qin et al. 2022; Tao et al. 2015; Yang et al. 2020; Yang et al. 2024; Yun et al. 2021). Despite its wide use in small-scale areas or specific forest types (Jurjević et al. 2020; Kwong and Fung 2020; Næsset and Økland 2002; Su et al. 2017; Yin et al. 2024), research gaps remain regarding large scale application of the tree-based approach. Moreover, the

absence of large-scale forest stand height metrics derived from the tree-based approach hinders the comparisons between tree-based and area-based methods for tree height estimation. Thus, to effectively implement sustainable management and development practices that balance conservation and human use needs, it is crucial to have comprehensive, timely, and accurate inventory and monitoring efforts for the height of forests at a national scale. (Coops et al. 2021; Liu et al. 2022; Travers-Smith et al. 2024). However, there are challenges associated with using close-range LiDAR to collect continuous large-scale forest

stand height observations considering the sparse coverage of LiDAR data and the associated costs.

To overcome the spatial discontinuity problem of close-range LiDAR samples on a large spatial scale, integration of multiple types of remote sensing data is a commonly used method for mapping wall-to-wall forest height (Huang et al. 2017; Lefsky 2010; Lefsky et al. 2005; Liu et al. 2022; Su et al. 2017; Travers-Smith et al. 2024). Current approaches typically rely on spatial interpolation and regression techniques. Spatial interpolation involves predicting values at unobserved locations based

on observed data points (Allard, 2013), taking advantage of spaceborne LiDAR data in wall-to-wall maps across extensive geographic areas with high resolution (Liu et al. 2022). For example, Liu et al. (2022) developed a spatial interpolation method to map China's forest canopy height by fusing GEDI, ICESat-2 ATLAS, and Sentinel-2 images. When it comes to forest stand mean height mapping, the spatial interpolation method may not be the most suitable method due to the rarity of forest stand mean heights in current spaceborne systems. In contrast, the regression strategy utilizes the continuous characteristics of optical

remote sensing, radar, and existing data products as predictors to construct non-linear mathematical models linking environmental factors with observations. For example, Su et al. (2017) used the random forest algorithm to model forest stand height in Sierra Nevada based on GLAS tree heights, optical imagery, topographic data and climate information. Travers-Smith et al. (2024) combined LiDAR and optical remote sensing products to map vegetation in high latitudes with a high overall accuracy. Currently, machine learning (ML) algorithms (Cheng et al. 2024a; Cheng et al. 2024b; Coops et al. 2021;

Matasci et al. 2018) and deep learning algorithms (Fayad et al. 2024; Liu et al. 2022) are the primary non-linear mathematical regression approaches for achieving large-scale continuous spatial forest attributes mapping. Compared to deep learning

algorithms, ML algorithms based on structured data are easier to implement and less computationally intensive. More importantly, in the context of mapping forest stand heights, feature engineering to compute relevant vegetation metrics is often preferred over utilizing raw radar and optical remote sensing data (Li et al. 2020; Potapov et al. 2021). Therefore, for estimation

the forest stand height through a tree-based approach at large scale, the regression strategy using ML algorithms provides a robust solution for integrating multi-source remote data for wall-to-wall forest stand height mapping.

Variations in forest types within different vegetation divisions on large spatial scales may also influence the accuracy of forest stand height mapping. One feasible solution is to develop specific model for each ecozone (Wu and Shi 2023). However, this approach may lead to noticeable boundary effects when estimating results based on multiple specific models. Therefore,

addressing how to adequately account for the spatial differences on a large spatial scale while using a single-global model is a problem that requires to be solved. The mixed-effects model, as demonstrated by Choi et al. (2024), offers a potential solution to simultaneously consider the heterogeneities of different regions. By integrating the mixed-effects model with ML, as proposed by Hu and Szymczak (2023), one can effectively leverage the strengths of both approaches to address complex data analysis challenges. This combination allows for the consideration of both random and fixed effects in the data while

harnessing the flexibility and non-linear mathematical regression of ML to better explain the complexity of the data.

In this study, our main objective is to map the national scale forest stand mean height across China through machine learning. We aim to address the challenges and potential for continuous mapping of tree height in heterogeneous forest ecosystem, through a tree-based approach. To train machine learning models, we have collected over 1117 km$^2$ of UAV LiDAR data across China. Subsequently, we mapped two forest stand mean height products at 30 m resolution: the arithmetic mean height

($h_a$) and the weighted mean height ($h_w$). Furthermore, we have validated the resulting forest stand mean height products by comparing them with field measurements and UAV LiDAR validation data. The national-scale maps of forest stand mean height produced in this study hold various applications, including estimating forest inventory, developing climate change mitigation strategies, monitoring changes in forest structure, and assessing wildlife habitat.

## 2 Material and methods

As illustrated in Figure 1, the workflow for this study involved four main steps: (1) Close-Range LiDAR data processing, including individual tree segmentation and forest stand mean height calculation, (2) Feature set construction, including multi-source remote sensing data and ancillary data processing, (3) ML-based mixed-effects (MLME) modeling, (4) Mapping of wall-to-wall forest stand mean height across China, including accuracy assessment and uncertainty analysis.

Earth System Science Data Discussions Open Access



**Figure 1: Workflow adopted for the modeling and mapping forest stand mean heights ($h_a$ and $h_w$) at 30 m resolution across the China's forest. Publisher's remark: please note that the above figure contains disputed territories.**

## 2.1 Close-range LiDAR data

The close-range LiDAR data used in this study were collected since 2015, covering various types of vegetation divisions across China, excluding the Qinghai-Tibet Plateau alpine vegetation divisions (Table 1, 2 and Fig. 2). The UAV LiDAR system was utilized in this study, resulting in a total data volume of 400 TB and covering an area of 1117.76 km$^2$ (Table 2 and Fig. 2). These data serve as the foundation for creating the forest stand mean heights sample set. The LiDAR data underwent initial processing using LiDAR360 software (V 6.0, www.lidar360.com), which included resampling, denoising, ground point classification, and normalization. The processed LiDAR data were then partitioned into $30 \times 30$ m grids, representing plots or stands of forest. In total, there were 610,342 plots created. To identify individual trees with height attribute within $30 \times 30$ m plots, the individual tree segmentation algorithm (Li et al. 2012) was introduced through the LiDAR360 software that designed



for Manned/UAV LiDAR data. Manual visual inspection was conducted for each plot to determine the optimal parameters for individual tree segmentation within the LiDAR360 software, ensuring more precise and reliable results. The dataset was used to train and validated the maps of forest stand mean heights in this study.

**Table 1.** LiDAR sensor parameter information during 2015 and 2023.

| LiDAR sensor | MAX points per second(pts/s) | Wave length | Range accuracy | Product |
|---|---|---|---|---|
| Pandar40P | 1,440,000(dual return) | 905 nm | 2 cm | LiAir 220N |
| Riegl mini VUX-1UAV | 100,000 | Near Infrared | 1.5 cm | LiAir 250 |
| XT32M2X | 1,920,000(triple return) | 905nm | 1.0 cm | LiAir 300 |
| Riegl VUX-1LR-22 | 1,500,000 | Near Infrared | 1.5 cm | LiAir D1350 |
| Riegl VUX-1LR-22 | 1,500,000 | Near Infrared | 1.5 cm | LiAir E1350 |
| Riegl VUX-120-23 | 2,000,000 | Near Infrared | 1.0 cm | LiAir E1500 |
| Riegl VUX-1LR-22 | 1,500,000 | Near Infrared | 1.5 cm | LiAir H2.0 |
| Riegl VUX-120-23 | 2,000,000 | Near Infrared | 1.0 cm | LiAir H1500 |
| Riegl VUX-1LR-22 | 1,500,000 | Near Infrared | 1.5 cm | LiAir H1800 |
| Livox Horizon | 240,000 | 905 nm | 2.0 cm | LiAir VH |
| Livox Mid-40 | 100,000 | 905 nm | 2.0 cm | LiAir V |
| Livox AVIA | 720,000(triple return) | 905 nm | 2.0 cm | LiAir V70 |
| Livox AVIA | 720,000(triple return) | 905 nm | 2.0 cm | LiAir VH2 |
| Velodyne's Puck | 600,000(dual Return ) | 903 nm | 3.0 cm | LiAir 50N |
| Riegl VUX-1LR-22 | 1,500,000 | Near Infrared | 1.5 cm | LiHawk |


**Table 2.** Summary of the collected UAV LiDAR data grouped in eight vegetation divisions across China.

| Vegetation division | Total area(km$^2$) | Proportion of forest area covered by drone lidar data ($\times 10^{-5}$) |
|---|---|---|
| Temperate desert (TD) | 0.18 | 0.75 |
| Temperate steppe (TS) | 76.04 | 98.16 |
| Subtropical evergreen broadleaf forest (SE) | 755.91 | 54.28 |
| Tropical monsoon forest-rainforest (TM) | 65.42 | 39.64 |
| Warm temperate deciduous broadleaf forest (WT) | 163.97 | 74.92 |
| Temperate needleleaf-broadleaf mixed forest (TN) | 53.61 | 19.46 |
| Cold temperate needleleaf forest (CT) | 2.63 | 1.16 |

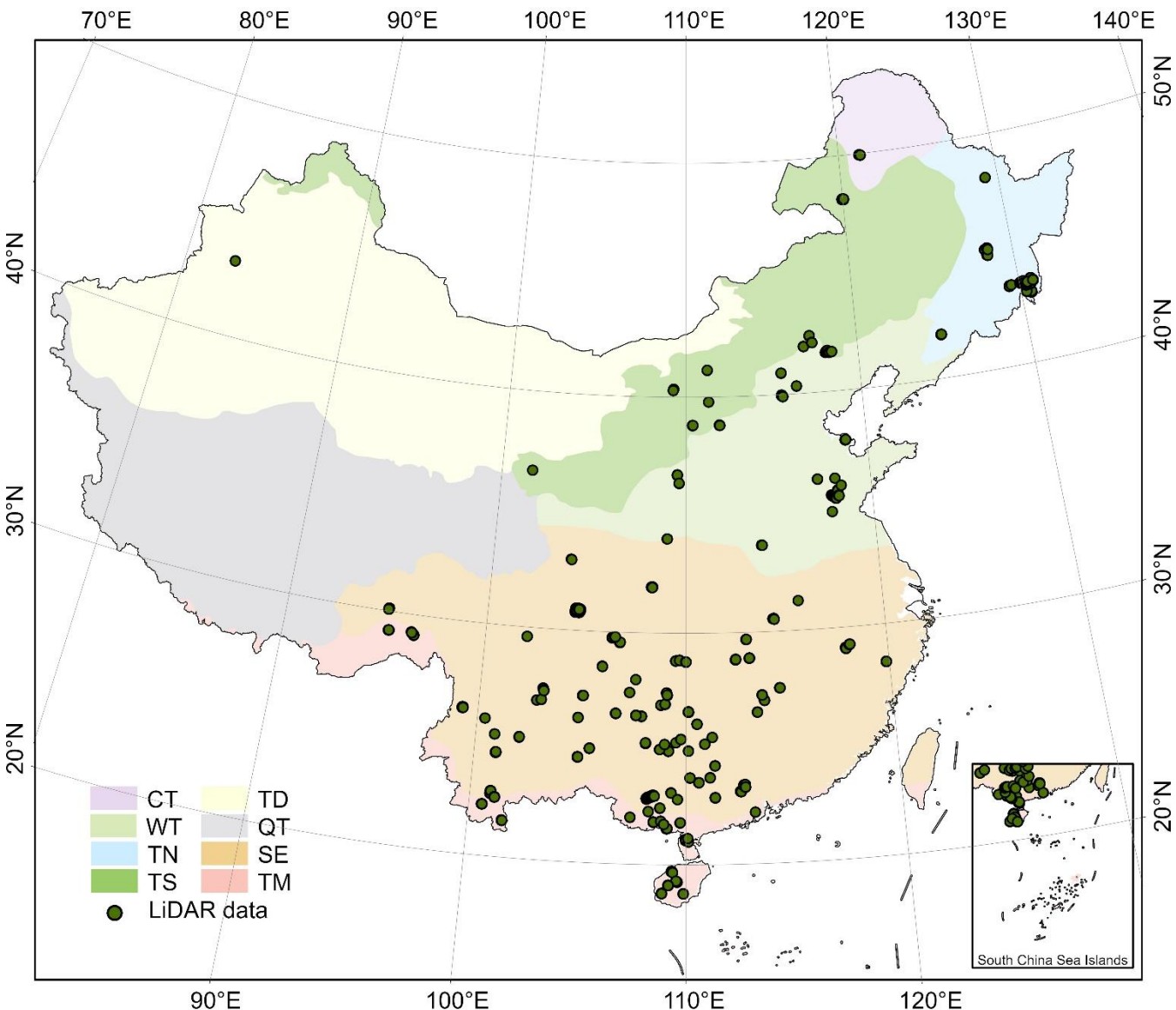

**Figure 2: Spatial distribution of the close-range LiDAR data used in this study. CT, TN, TS, TD, WT, QT, SE, and TM represent the vegetation divisions of cold temperate needleleaf forest, temperate needleleaf-broadleaf mixed forest, temperate steppe, temperate desert, warm temperate deciduous-broadleaf forest, Qinghai-Tibet Plateau alpine vegetation, subtropical evergreen broadleaf forest, and tropical monsoon forest-rainforest, respectively. Publisher's remark: please note that the above figure contains disputed territories.**

## 2.2 Field data

Field samples were collected for weighted mean height calculation (see Section 2.3) and product validation (see Section 2.5). From 2019 to 2023, a total of 294 plots were collected in six provinces of China. All of the plots achieved decimeter-level positioning accuracy, and each plot covered an area Larger than 400 m². The center position and information of individual





trees in each plot, including DBH (DBH > 5 cm) and tree height were recorded in each plot. However, it is worth noting that there may exists the time discrepancy between the field surveys and LiDAR data acquisition. Additionally, variability may be introduced due to multiple field surveyors and region-specific adjustments to measurement tools.

**2.3 Tree-based approach for calculating Forest stand mean heights**

Forest stand mean heights, including $h_a$ and $h_w$, were calculated using a tree-based approach in this study (Laar and Akça 2007; Nakai et al. 2010). While accurately segmenting individual trees based on LiDAR data might be challenging due to missing smaller trees and trees obscured by understory vegetation (Li et al. 2012; Tao et al. 2015; Yang et al. 2024), leading to some under-segmentation, this step remains crucial for modeling and mapping forest stand mean heights. Therefore, in this

study, we only considered the parameter extraction of successfully segmented trees when using the tree-based approach for the inverting and mapping forest stand mean heights. The segmented individual tree results obtained from close-range LiDAR data contained latitude, longitude, and tree height information for each tree.

**2.3.1 Arithmetic mean height ($h_a$)**

The arithmetic mean height ($h_a$) is calculated as the average of the tree heights of all trees obtained from the $30 \times 30$ m plots

basing on our UAV LiDAR data. It is calculated as following:

$$h_a = \frac{\Sigma_{i=1}^{n} h_i}{n} \tag{1}$$

where $h_i$ is the height of *i-th* tree (usually with a threshold of $h_i \geq 2.0$ m), *n* is the number of trees within the grid.

**2.3.2 Weighted mean height ($h_w$)**

Weighted mean height ($h_w$) is calculated by using the basal area as the weight for determining the forest stand mean height (Laar and Akça 2007; Lorey 1878; Masaka et al. 2013). It is calculated as following:

$$h_w = \frac{\Sigma_{i=1}^{n} w_i h_i}{\Sigma_{i=1}^{n} w_i} \tag{2}$$

where $h_w$ represents the weighted mean height (m), $h_i$ is the height of *i-th* tree (usually with a threshold of $h_i \geq 2.0$ m), $w_i$ is the weight (basal area) of *i-th* tree, *n* is the number of trees within the plot or stand.

Considering the limitations in obtaining basal area of individual trees within plots using UAV LiDAR data, the basal area cannot be used as a weight for calculating $h_w$ in this study. According to Næsset (1997), since the basal area of a tree is closely correlated to its height, the value of an individual tree height can be weighted by its tree height or even by the square of its

height. Therefore, in this study, we adopted Næsset (1997) method to calculate $h_w$, taking the tree height ($w_1$) and square of tree height ($w_2$) as two alternative weights. The results were compared with the $h_w$ weighted based on basal area, the better-performing one was selected as the final weight in this study. To determine the optimal weight, theoretical growth equations (Supplementary Table S1) for stand age and Lorey's mean height ($h_L$, based on basal area) were constructed employing



national forest inventory data (Cheng et al. 2024a). The optimal logistic model was selected as the stand age - $h_L$ model

(Supplementary Table S2 and S3). Then, close-range LiDAR data and forest age data from Cheng et al. (2024a) were combined

with the selected model to calculate $\hat{h}_L$ weighted by basal area. In addition, $h_{w1}$ and $h_{w2}$ were calculated based on weights of

$w_1$ and $w_2$, respectively. The errors between $\hat{h}_L$ and the two weighted mean heights ($h_{w1}$ and $h_{w2}$) were derived and the one

with smaller error were selected as the weighted mean height $h_w$ for this study (Supplementary Table S4). Finally, to confirm

the accuracy of the selected weighted $h_w$ from the second step, it was compared with the basal area-weighted $h_L$ of 199 sample

plots, calculated using integrated LiDAR and field data (including manually measured DBH and LiDAR-measured tree heights

for each plot). $h_w$ weighted by $w_2$ showed a strong correlation with $h_L$, with $r = 0.92$, RMSE = 1.8 m and MAE = 1.0 m (Fig.

3 and Supplementary Table S4). Thus, in this study, $w_2$ was used as the weight in the calculation of $h_w$.

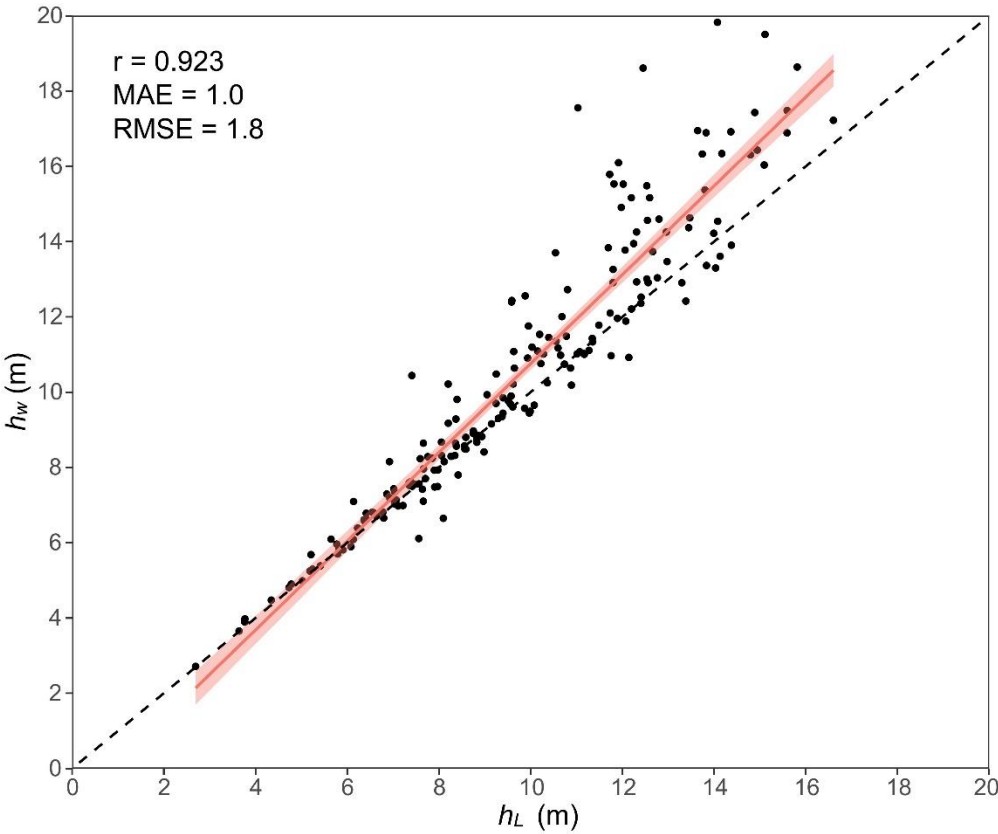

**Figure 3: The scatter plot for correlation analysis between $h_w$ (weighted by its square of tree height) and $h_L$**

**2.4 Ancillary data**

In order to invert the forest stand mean height across China, 30 geospatial features were derived from multi-source remote

sensing data. These those features were grouped into five categories: climatic, edaphic, topographic, vegetative, and SAR data

as shown in Table 3. Climate data were obtained from WorldClim 2.1 (www.worldclim.org), which provides 19 bioclimatic





variables including temperature and precipitation at a 30 arc-second resolution. The Harmonized World Soil Database (HWSD)
V1.2, with a resolution of 30 arc-second, was used to extract soil factors, including soil type and soil texture, to construct the
tree density estimation model. The Shuttle Radar Topography Mission (SRTM) V3 product (SRTM Plus) provides global
digital elevation data at 1 arc-second resolution and was used to extract topographic features. Three topographic features, i.e.,
elevation, slope, and aspect, were calculated using the Python *ee* package in this study (Table 3). The vegetative index used in
this study is the Landsat-based annual maximum composite Normalized Digital Vegetation Index (NDVI) obtained from the
Google Earth Engine (GEE). The SAR data, including VV and VH, were derived from Sentinel-1. The planted and natural
forest map was used to separate natural forests from planted ones (Cheng et al. 2024b). Additionally, the forest age map for
China in 2020 (Cheng et al. 2024a) was utilized. For consistency, all geospatial features were resampled to a 30 m resolution
using the nearest resampling method.

**Table 3.** Descriptions of multi-source remote sensing variables used to map China's forest stand mean heights product.

| Data type | Data source | Resolution | Time | Variables |
|---|---|---|---|---|
| Forest mask | Planted and natural forest map (Cheng et al.,2023b) | 30m | 2020 | Planted and natural forest |
| Forest age | Forest age map (Cheng et al.,2023a) | 30m | 2020 | Forest age |
| SAR data | Sentinel-1 | 30m | 2015–2020 | VV, VH |
| Landsat data | Landsat | 30m | 2015–2020 | NDVI_MAX |
| Climate data | WorldClim V2.1 | 30 arc-second | 1970–2000 | BIO1-BIO19 |
| Soil data | Harmonized World Soil Database V1.2 | 30 arc-second | 2007–2009 | SU_SYM90, REF_DEPTH, T_TEXTURE |
| Topographic data | SRTM DEM | 30 m | 2000 | Elevation, Slope, Aspect |

**2.5 ML-based mixed-effects model to map wall-to-wall forest stand mean height of China**

The modeling data of the study have a hierarchical structure, including eight vegetation divisions, each hosting its specific
plots. It is noteworthy that plots within the same vegetation division are not independent, and they exhibit notable heterogeneity
when compared across various vegetation divisions. To address the problems, we proposed a novel machine learning-based
mixed-effects model framework. It integrates machine learning algorithms with mixed-effects model methodology, providing
a powerful and flexible solution for modeling heterogeneous hierarchical data. This method not only enhances the accuracy of
modeling, but also provides in-depth results for studies with complex data structures.



### 2.5.1 ML-based mixed-effects model (MLME)

We assumed there are $n$ vegetation divisions, each containing forest stand mean height measurements taken at $m$ spatial points. These spatial points correspond to $30 \times 30$ m plots in this study. We denote the $t$-th ($t = 1, 2, ..., m$) plot measurement in the $i$-th ($i = 1, 2, ..., n$) vegetation division as $h_{it}$. The means, variances, and correlations of the forest stand mean height within the $n$ vegetation divisions in each given plot, can be expressed as.

$$\mu = \begin{bmatrix} \mu_1 \\ \mu_2 \\ \vdots \\ \mu_m \end{bmatrix} \quad \sigma^2 = \begin{bmatrix} \sigma_1{}^2 \\ \sigma_2{}^2 \\ \vdots \\ \sigma_m{}^2 \end{bmatrix} \quad CORR = \begin{bmatrix} 1 & \rho_{12} \cdots & \rho_{1m} \\ \rho_{21} & 1 \cdots & \rho_{2m} \\ \vdots & \vdots \ \vdots & \vdots \\ \rho_{m1} & \rho_{m2} \cdots & 1 \end{bmatrix} \tag{3}$$

A non-linear mean function was chosen.

$$\mu_t \cong f(X_t; \beta), t = 1, 2, ..., \text{m} \tag{4}$$

where $\beta$ is the parameter vector, $X_t$ is predictor variables at plot $t$.

With the addition of certain random effects to the base model of Equation. 4, a nonlinear mixed-effects model of general form is obtained.

$$h_i = f(X_i; \beta, b_i) + \varepsilon_i, i = 1, 2, ..., n \tag{5}$$

where $f$ denotes the forest stand mean height function, $h_i$ is the forest stand mean height at the $i$-th vegetation division, $\beta$ is the population parameter vector that is common to all vegetation divisions, and $b_i$ is the random-effect vector specific to the $i$-th vegetation division, the $\varepsilon_i$ is the error term.

To utilize linear prediction, we incorporate the random effects $b_i$ to linearize model (Equation 5) with respect to the random effects as shown in Equation 6. This model, linear in random effects, is termed a nonlinear marginal model (Demidenko 2013; Wang et al. 2023).

$$h_i \cong f(X_i; \beta) + Z_i b_i + \varepsilon_i, i = 1, 2, ..., n \tag{6}$$

where $Z_i$ is the matrix of first-order derivatives with respect to the random effects.

The model described in Equation 6 is decomposed into two distinct components: one related to fixed effects and the other related to random effects. For the fixed effect functions, we employed ML models, proposing the MLME model as described in Equation 7.

$$h_i \cong \beta_0 + \beta_1 ML(X_i) + Z_i b_i + \varepsilon_i, i = 1, 2, ..., n \tag{7}$$





where $ML(X_i)$ represents the predicted value from the ML of each plot $t$ (= 1, 2, ..., $m$) within the mixed-effects model for vegetation division $i$ (= 1, 2, ..., $n$). $\beta_0$ denotes the intercept coefficient and $\beta_1$ is the coefficient for the vector of predicted value for $ML(X_i)$. By using the predicted values from ML model, the influence of covariates is integrated into the mixed-effects model, thereby accounting for the specific forest stand mean height measurement within vegetation divisions.

**2.5.2 Mapping wall-to-wall forest stand mean heights across China**

According to workflow outlined in section 2.4.1, our work began with builting the ML models to determine the fixed effects. We employed *PyCaret,* an open-source, low-code ML library in Python (https://pycaret.org), which integrates various popular ML libraries and frameworks, to select ML algorithms for forest stand mean height estimation. Four ML algorithms, including random forest (RF), eXtreme Gradient Boosting (XGBoost), Light Gradient Boosting Machine (LightGBM), and Categorical
Boosting (CatBoost), demonstrated superior performance (Supplementary Table S5). Then, the Bayesian optimization was employed for hyperparameter tuning (Mekruksavanich et al. 2022), using *Optuna*, an open-source framework for hyperparameter optimization, to automate the search for hyperparameters (Akiba et al. 2019) (Supplementary Table S6 and S7). Next, the parameters of MLME were estimated using a two-step procedure. First, the forest stand mean height models were trained using the four ML algorithms to obtain $h_a$ and $h_w$. It was found that LightGBM demonstrate the optimal
performance and was selected to map $h_a$ and $h_w$ across China (Supplementary Table S8). Secondly, Equation 7 in section 2.4.1 was applied to derive vegetation division-specific estimates of heights based on the forest stand mean height obtained from these best LightGBM models. Specifically, the *lmer* function from the *lme4* package in the R language (version 4.3.0) was used for maximum likelihood estimation during model fitting.

**2.5.3 Accuracy assessment**

The original dataset was randomly split into two groups: 2/3 for the training set, and the remaining for the validation set. Several statistical indicators, including coefficient of determination ($R^2$), root mean square error (RMSE), mean square error (MSE), and mean absolute error (MAE) were calculated to evaluate the performance of the ML model in this research. Additionally, Akaike information criterion (AIC) and Bayesian information criterion (BIC) were employed to compare the accuracy and generalization of the parameter estimation results of the MLME model. The Intraclass Correlation Coefficient
(ICC) is a statistical measure used to quantify the reliability or agreement of measurements made by multiple plot observers measuring the same vegetation division. These statistical indicators are defined as follows:

$$R^2 = 1 - \frac{\sum_{i=1}^{n}(y_i - \hat{y}_i)^2}{\sum_{i=1}^{n}(y_i - \overline{y_i})^2} \tag{8}$$

$$MSE = \frac{1}{n}\sum_{i=1}^{n}(y_i - \hat{y}_i)^2 \tag{9}$$





$$RMSE = \sqrt{\frac{1}{n}\sum_{i=1}^{n}(y_i - \hat{y}_i)^2} \tag{10}$$

$$MAE = \frac{1}{n}\sum_{i=1}^{n}\left|y_i - \hat{y}_i\right| \tag{11}$$

$$ICC = \frac{\hat{\sigma}_b^{\ 2}}{\hat{\sigma}_b^{\ 2} + \hat{\sigma}^2} \tag{12}$$

$$AIC = -2 * \ln(L) + 2K \tag{13}$$

$$BIC = -K * ln(L) + Kln(n) \tag{14}$$

where $y_i$ represents the observed value for the *i-th* analytic tree; $\hat{y}_i$ is the predicted value of *i-th* observed value; *n* is the number of trees, $\overline{y_i}$ is the mean value for the observed values, $\hat{\sigma}_b^{\ 2}$ is random intercept variance,$\hat{\sigma}^2$ is residual variance, *L* is the value of log-likelihood, *K* is the number of parameters in the model.

## 2.6 Uncertainty analysis

The total uncertainty of forest stand mean heights at pixel level is divided into three independent terms, each terms reported as a percentage of relative uncertainty, following Saatchi et al. (2011):

*Measurement error* ($\varepsilon_{measurement}$) is associated with the accuracy of individual tree segmentation from close-range LiDAR data. This study employs (1−F1 score) value as a measure of the percentage of relative uncertainty. The overall F1 score value averaged across all the testing plots was 0.90 with a corresponding measurement uncertainty of 10% (Li et al. 2012).

*Product error* ($\varepsilon_{product}$) refers to errors in estimating forest age (Cheng et al. 2024a) from forest structure attributes maps. It was estimated from the relations developed from calibration plots. The relative uncertainty of forest age for each pixel is calculated using Equation 15, and then the maximum of mean relative error of forest age is calculated using Equation 16 (Supplementary Note S1).

$$\varepsilon_i = \frac{RMSE}{\hat{y}_i} \tag{15}$$

$$\bar{\varepsilon}_{max} = \frac{RMSE}{\bar{y}} \times 100\% \tag{16}$$

where *RMSE* is the root mean squared error of the validation set for the forest age product, $\hat{y}_i$ represents the predicted value of the *i-th* observed value.

*Prediction error* ($\varepsilon_{prediction}$) includes both the sampling error associated with the representativeness of the training data relative to the actual spatial distribution of stand height, as well as the model predictions. The relative uncertainty for each pixel is calculated as:



$$\varepsilon_{prediction} = \frac{RMSE_h}{\hat{h}_i} \tag{17}$$

where $RMSE_h$ is the root mean squared error of the validation set, $\hat{h}_i$ represents the predicted stand height value of the *i-th*

observed stand height value.

Finally, we propagated the errors through the entire process by assuming all errors were independent and random. The

uncertainty in estimating stand mean height ($\varepsilon_h$) was quantified using:

$$\varepsilon_{h_i} = \sqrt{\varepsilon_{measurement}^2 + \varepsilon_{product}^2 + \varepsilon_{prediction}^2} \tag{18}$$

To calculate the national-level uncertainty, we sum the errors from all pixels using:

$$\varepsilon_{national} = \frac{\sqrt{\sum_{i=1}^{N}(\hat{h}_i \varepsilon_{h_i})^2}}{\sum_{i=1}^{N} \hat{h}_i} \tag{19}$$

where $N$ is the number of pixels within the national boundary, and $\hat{h}_i$ and $\varepsilon_{h_i}$ are the stand height and its relative uncertainty

at pixel *i*, respectively.

## 3 Results

### 3.1 UAV LiDAR-derived arithmetic mean height ($h_a$) and weighted mean height ($h_w$) across China

The UAV LiDAR-derived $h_a$ and $h_w$ varied across seven vegetation divisions (excluding Qinghai-Tibet Plateau alpine

vegetation), However, as illustrated in Figure.4, the rankings of $h_a$ and $h_w$ remained consistent across these vegetation

divisions. The tallest LiDAR-derived $h_a$ and $h_w$ were recorded in the tropical monsoon forest-rainforest, measuring 68.49 m

and 69.67 m, respectively. In contrast, the highest average LiDAR-derived $h_a$ and $h_w$ were observed in the temperate desert,

with values of 30.55 ± 6,94 m and 37.28 ± 5.58 m, respectively (Fig. 4). The range of UAV LiDAR-derived $h_a$ and $h_w$

distribution was widest in the tropical monsoon forest-rainforest, with standard deviations of 8.00 m and 9.15m, respectively

(Fig. 4). While the lowest standard deviations of $h_a$ and $h_w$ were recorded in the cold temperate needleleaf forest, with values

of 2.17m and 1.94 m, respectively (Fig. 4). These UAV LiDAR-derived $h_a$ and $h_w$ provided a concrete foundation for training

and validating the tree-based approach for mapping wall-to-wall $h_a$ and $h_w$ of China's forest.

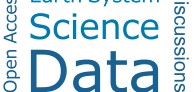





**Figure 4: Histograms of UAV LiDAR-derived $h_a$ and $h_w$ across vegetation zone and the overall dataset. Mean and SD represent the mean and standard deviation of UAV LiDAR-derived $h_a$ and $h_w$, with N represents the number of training sample plots (size =30 × 30 m). CT, TN, TS, TD, WT, SE, and TM represent the vegetation divisions of cold temperate needleleaf forest, temperate needleleaf-broadleaf mixed forest, temperate steppe, temperate desert, warm temperate deciduous-broadleaf forest, subtropical evergreen broadleaf forest, and tropical monsoon forest-rainforest, respectively.**





## 3.2 Prediction accuracy of MLME model

Based on the construction and optimization process of ML model (see Supplementary Table S6 and S7), the MLME models simultaneously account for the interaction between covariates (the predicted $h_a$ and $h_w$ from the ML) and the grouping variable (eight vegetation divisions). These models, as shown in Table 3 and Table 4, reveal different predictions of $h_a$ and $h_w$ across vegetation divisions. This study suggests that discrepancies in estimating $h_a$ and $h_w$ across China primarily stem from variations in vegetation divisions, as evidenced by ICC values of 0.581 and 0.693 for $h_a$ and $h_w$, respectively (Table 4).

These values indicated that approximately 58.1% and 69.3% of the total variance are attributable to variations between different vegetation divisions for estimating $h_a$ and $h_w$, respectively (Table 4).

**Table 4.** Parameter estimates and fitting statistics of MLME models for vegetation divisions.

|  | Vegetation division | $h_a$ | | $h_w$ | |
|---|---|---|---|---|---|
|  |  | b | $\beta$ | b | $\beta$ |
| Parameter estimates | CT | 0.182 | 0.989 | -0.249 | 1.016 |
|  | WT | -0.108 | 1.007 | -0.249 | 1.018 |
|  | TM | 0.133 | 0.992 | -0.249 | 1.011 |
|  | TS | 0.111 | 0.993 | -0.249 | 1.015 |
|  | TD | 0.56 | 0.964 | -0.249 | 1.016 |
|  | TN | 0.284 | 0.982 | -0.249 | 1.016 |
|  | SE | -0.042 | 1.003 | -0.249 | 1.021 |
| Fitting statistics | ICC | 0.581 | | 0.693 | |
|  | AIC | 962107 | | 1008650 | |
|  | BIC | 962168 | | 1008711 | |
|  | logLik | -481048 | | -504319 | |
|  | Pr (>Chisq) | 3.37E-05*** | | 9.29 E-06*** | |

Notes: $h_a$ is arithmetic mean height; $h_w$ is weighted mean height; CT is cold temperate needleleaf forest, TN is temperate needleleaf-broadleaf mixed forest, TM is tropical monsoon forest-rainforest, TS is temperate steppe, TD is temperate desert,

WT is warm temperate deciduous-broadleaf forest, SE is subtropical evergreen broadleaf forest.

Our results demonstrated high accuracy in estimating $h_a$ and $h_w$ across China's forests, with $R^2 > 0.82$, RMSE < 2.6 m and MAE < 1.9 m for $h_a$ and $R^2 > 0.78$, RMSE < 2.9 m and MAE < 2.1 m for $h_w$ (Table 5 and Fig. 5). It can be seen that the accuracy of $h_a$ consistently surpasses that of $h_w$, regardless of whether ML or MLME methods were employed. As displayed

in Table 5 and Fig. 5, MLME models, which integrated vegetation divisions as variables, showed slightly superior performance in estimating $h_a$ and $h_w$ compared to ML models, once again highlighting the impact of incorporating vegetation divisions on

the estimation results. Additionally, incorporating vegetation divisions improved the accuracy of $h_w$ estimation slightly when compared to that of $h_a$, This may be due to the stronger influence of vegetation divisions on $h_w$ (ICC=0.693) (Table 4 and 5). Fig. 5 provides comparisons of estimated $h_a$ and $h_w$ against observed LiDAR validation data based on MLME. Excellent

consistency can be found between estimated and validation results. High correlations are presented in the $h_a$ MLME model (r=0.906), as well as $h_w$ MLME model (r=0.889). Specifically, the $h_a$ tended to be slightly overestimated for lower measured mean height, while no biased estimation was observed for $h_w$.

**Table 5** Comparative prediction accuracy of $h_a$ and $h_w$ models with ML and MLME.

| Model | Method | R$^2$ | RMSE | MAE |
|---|---|---|---|---|
| $h_a$ | ML | 0.820899 | 2.5744 | 1.8581 |
| | MLME | 0.820940 | 2.5741 | 1.8567 |
| $h_w$ | ML | 0.7892 | 2.8876 | 2.0940 |
| | MLME | 0.7895 | 2.8859 | 2.0888 |

Notes: $h_a$ is arithmetic mean height; $h_w$ is weighted mean height

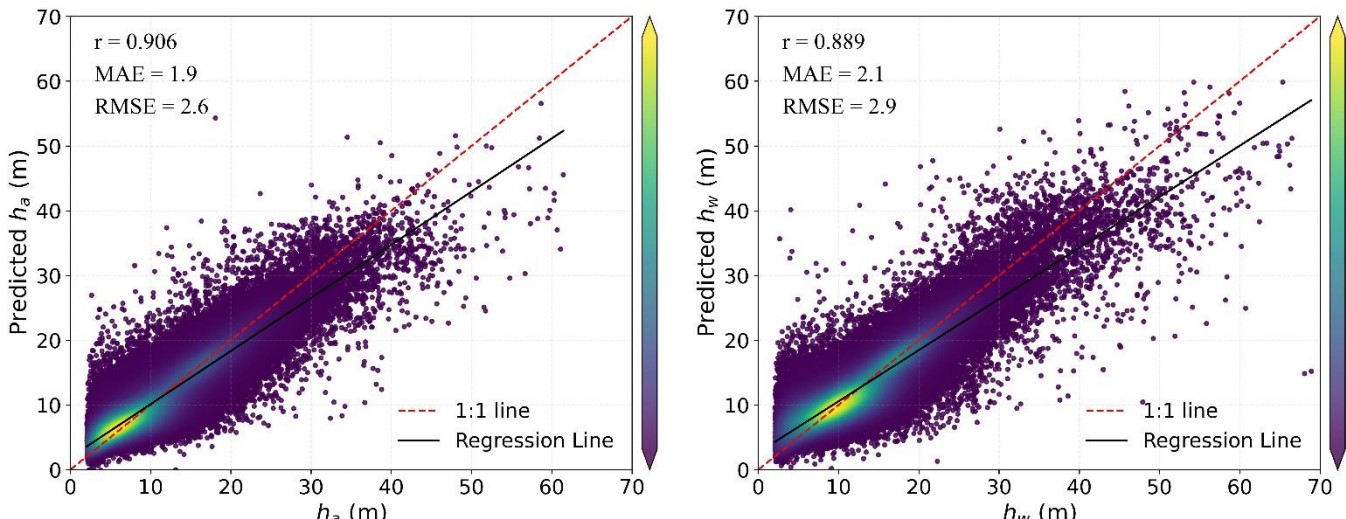


**Figure 5: Accuracy assessment of the MLME-derived $h_a$ and $h_w$ when compared with LiDAR validation data.**

### 3.3 Wall-to-wall arithmetic mean height ($h_a$) and weighted mean height ($h_w$) of China

We applied MLME models to map $h_a$ and $h_w$ at a 30 m resolution across China's forests, as shown in Fig. 6 and Fig. 7, respectively. Overall, the mean value of $h_w$ was 13.3 ± 3.3 m (mean value ± SD) for pixels, which is higher than that of $h_a$

(11.3 ± 2.9 m) per pixels. Geographically, our $h_a$ and $h_w$ maps exhibited similar patterns, with forests in the southwestern ($h_a$ = 12.05 ± 3.27 m, $h_w$ = 14.13 ± 3.85 m), northeastern ($h_a$ = 12.05 ± 3.20 m, $h_w$ = 14.74 ± 3.39 m), and southeastern ($h_a$ =10.98 ± 2.67 m, $h_w$ = 12.67 ± 2.60 m) part of China being relatively taller than those in the southern ($h_a$ = 10.69 ± 2.39 m, $h_w$ = 12.56 ± 2.39 m), northern ($h_a$ = 10.28 ± 2.44 m, $h_w$ = 11.80 ±2.43 m) and northwestern ($h_a$ = 10.63 ± 2.53 m, $h_w$ =





12.59 ± 2.87 m) regions (Figs. 6 and 7). Despite considerable variations in $h_a$ and $h_w$ among forests in different provinces

(Fig. 8), a consistent trend indicated that $h_a$ was generally lower than $h_w$ across all provinces, except for Tianjin ($h_a = 8.78 ±$ 1.64 m VS. $h_w = 8.34± 1.64$ m) (Fig. 8). Notably, Jilin exhibited the highest $h_a$ value (13.41 ± 2.92 m), closely followed by Tibet (13.40 ± 3.60 m), Anhui (12.96 ± 2.65 m), Xinjiang (12.39 ± 4.01 m), and Sichuan (12.21 ± 3.02 m). Conversely, Shanghai recorded the highest $h_w$ value (17.17 ± 2.02 m), with Jilin (17.12 ± 2.87 m), Xinjiang (16.93 ± 2.54 m), Tibet (16.21 ± 4.73 m), and Yunnan (15.03 ± 4.02 m) trailing closely behind. Tibet, Xinjiang and Yunnan showed greater variability in

both $h_a$ and $h_w$ than in other provinces.



**Figure 6: The forest arithmetic mean height ($h_a$) of China derived from the tree-based approach at 30 m resolution for 2020. Publisher's remark: please note that the above figure contains disputed territories.**





Figure 7: The forest weighted mean height ($h_w$) of China derived from tree-based approach at 30 m resolution for 2020. Publisher's remark: please note that the above figure contains disputed territories.



**Figure 8: Province-level analysis of $h_a$ and $h_w$ estimations derived from tree-based approach. The black circle is the mean value, the solid line in box is median value for each province.**

The estimations for $h_a$ and $h_w$ conformed to a normal distribution across eight vegetation divisions in China, with there are some notable differences between the two as shown in Fig. 9. The highest recorded values for $h_a$ and $h_w$ values were 94 m and 96 m, respectively, observed in subtropical evergreen broadleaf forest. For $h_a$, the temperate needleleaf-broadleaf mixed forest exhibited the tallest forest height (median = 13 m, mean = 13.03 m), followed by tropical monsoon forest-rainforest (median = 13 m, mean = 12.95 m), Qinghai-Tibet Plateau alpine vegetation (median = 12 m, mean = 12.38 m), temperate

desert (median = 11 m, mean = 11.83 m), subtropical evergreen broadleaf forest (median = 11 m, mean = 11.14 m), temperate steppe (median = 11 m, mean = 10.78 m), cold temperate needleleaf forest (median = 11 m, mean = 10.75 m) and warm



temperate deciduous-broadleaf forest (median = 9 m, mean = 9.34 m). For $h_w$, tropical monsoon forest-rainforest had the tallest forest height (median = 16 m, mean = 16.41 m), followed by temperate needleleaf-broadleaf mixed forest (median = 16 m, mean = 15.96 m), temperate desert (median = 16 m, mean = 15.44 m), cold temperate needleleaf forest (median = 13 m, mean = 12.89 m), subtropical evergreen broadleaf forest (median = 12 m, mean = 12.84 m), temperate steppe (median = 13 m, mean = 12.80 m), Qinghai-Tibet Plateau alpine vegetation (median = 12 m, mean = 12.34 m) and warm temperate deciduous-broadleaf forest (median = 11 m, mean = 10.77 m). Compared to the ranges of $h_a$ distributions in different vegetation divisions, the ranges of $h_w$ distributions were wider as illustrated in Fig. 9. Specifically, the vegetation divisions with the greatest variations, the ranges of both $h_a$ and $h_w$ distributions were temperate desert (standard deviation = 4.09 m and 4.55 m of $h_a$ and $h_w$, respectively), tropical monsoon forest-rainforest (standard deviation = 3.71 m and 4.39 m of $h_a$ and $h_w$, respectively), and temperate needleleaf-broadleaf mixed forest (standard deviation = 3.10 m and 3.27 m of $h_a$ and $h_w$, respectively).





**Figure 9: Vegetation divisions-level analysis of the tree-based approach-derived $h_a$ and $h_w$ estimations. CT, TN, TS, TD, WT, QT, SE, and TM represent the vegetation divisions of cold temperate needleleaf forest, temperate needleleaf-broadleaf mixed forest, temperate steppe, temperate desert, warm temperate deciduous-broadleaf forest, Qinghai-Tibet Plateau alpine vegetation, subtropical evergreen broadleaf forest, and tropical monsoon forest-rainforest, respectively. The solid line in each box is median value.**

### 3.4 Uncertainty analysis

The uncertainties of $h_a$ (Fig. 10) and $h_w$ (Fig. 11) at the 30 m pixel level were quantified by integrating measurement error, product error, and prediction error of $h_a$ and $h_w$ maps. The overall uncertainty in mapping $h_a$ and $h_w$ at the pixel scale, averaged over all vegetation divisions, was estimated to be 23% and 21% respectively (Fig. 10 and Fig. 11). However, these uncertainties were not uniformly distributed across China. The uncertainty for $h_a$ ranged from 16% to 56%, while for $h_w$, it ranged from 16% to 59%. These variations depended on regional differences in forests, the quality of remote sensing imagery, and the sampling size and distribution of available field and LiDAR data. We further assessed the uncertainty around $h_a$ and $h_w$ estimates at national and regional scales by errors propagation. As the sample area increased, the relative errors decreased. The national estimations were found to be constrained to within 1% for $h_a$ ($4.30 \times 10^{-4}$ %) and $h_w$ ($4.12 \times 10^{-4}$ %).



**Figure 10:** Uncertainty analysis in the spatial distribution of forest arithmetic mean height ($h_a$) in each pixel. Publisher's remark: please note that the above figure contains disputed territories.

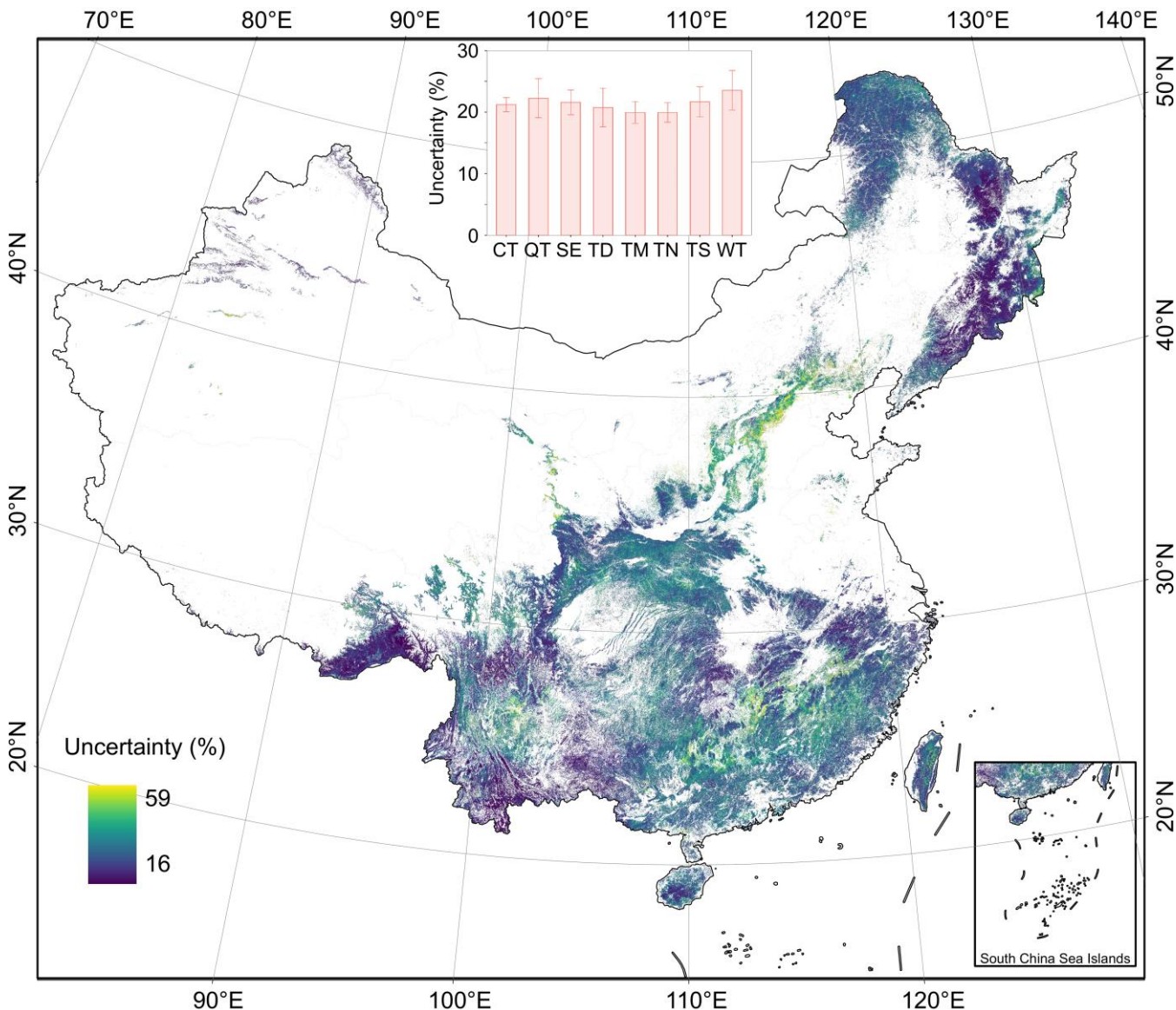

**Figure 11:** Uncertainty analysis in the spatial distribution of forest weighted mean height ($h_w$) in each pixel. Publisher's remark: please note that the above figure contains disputed territories.

## 4 Discussion

The national scale, continuous maps of arithmetic mean height ($h_a$) and weighted mean height ($h_w$) across China address the challenges of accurately estimating forest stand mean height using a tree-based approach. These maps provide critical datasets for forest sustainable management in China, including climate change mitigation (e.g., terrestrial carbon estimation) (Duncanson et al. 2022; Migliavacca et al. 2021), forest ecosystem assessment (Davies et al. 2017; Jucker et al. 2018; Li et al.





2020), and forest inventory practices (Fang et al. 2006; Fang et al. 2001; Travers-Smith et al. 2024; Xu et al. 2019). By leveraging high-point density, and high-precision close-range LiDAR data, spatially continuous maps that comply with the

definitions of $h_a$ and $h_w$ in forestry were generated. Validation results indicate that our method has a high accuracy (Figs. 5 and 12), demonstrating the potential for widespread application of tree-based forest stand mean height estimation at large-scale. Furthermore, our findings suggest that close-range LiDAR technology can enhance traditional forestry surveys by enabling rapid, accurate, large-scale, and cost-effective assessments.

Regarding forest stand mean height estimations ($h_a$ and $h_w$), distinctive methods and definitions of tree height from the

commonly used CHM are employed in this study. Despite these differences, previous studies indicate that there was a strong correlation between forest stand mean height and the CHM-derived height metrics from LiDAR point clouds. Finer scale analysis demonstrated that the 50%, 75%, 80%, 90%, and 95% percentile heights in the CHM model can serve as the optimal variables for linear regression prediction of Lorey's weighted height across different studies (Li et al. 2022; Liu et al. 2018; Pang et al. 2008). Conversely, for predicting arithmetic mean height, different optimal variables might be employed, including

the 30%, 60%, and 70% percentile heights (Jensen and Mathews 2016; Li et al. 2022). The above-mentioned differences resulting from variations in geographic regions and tree species highlight the challenges in indirectly estimating $h_a$ and $h_w$ using the CHM (Yin et al. 2024). Spaceborne LiDAR and multi-source remote sensing data have been widely used for estimating forest canopy height on national/global scale through area-based approaches (Coops et al. 2021; Liu et al. 2022; Travers-Smith et al. 2024), including the 98% percentile height (Lang et al. 2023; Liu et al. 2022), 95% percentile height

(Potapov et al. 2021), 100% percentile height (Ni et al. 2015; Simard et al. 2011), and 90% percentile height canopy height products. However, these results are more align with the maximum tree height which may be numerically closer to the forest dominant/top height (Li et al. 2023), rather than the forestry definitions of forest stand mean height (Laar and Akça 2007; Masaka et al. 2013). In forestry, the dominant tree height is widely recognized as a key factor in explaining forest site productivity (Vanclay 1992; Vatandaslar et al. 2023; Woods et al. 2011), while stand mean heights are crucial for calculating

forest volume and carbon storage capacity (Xu et al. 2019). As the demand for accurate tree height estimation grows, our study aligns with forestry definitions and needs. Taking advantages of extensive high-precision UAV LiDAR data, we mapped the national-scale forest $h_a$ and $h_w$ data products for China. Among them, $h_a$ can efficiently and accurately assesses the stand mean height in even-aged stands (e.g., planted forest), while $h_w$ is a valuable parameter for representing the mean height in uneven-aged forest stands (e.g., natural forest). Our maps underscore the importance of using appropriate tree height definitions

and methodologies tailored to meet the specific requirements of forestry management and ecological research.

A significant challenge of using a tree-based approach to calculated Lorey's weighted height from UAV LiDAR data lies in the difficulty of obtaining DBH information. Consequently, the traditional Lorey's weighted height calculation method is not feasible in this study. To address this problem, the regression height of the quadratic mean diameter method, which estimate stand mean height by corresponding to the tree height with average DBH on the height-diameter curve, served as a simplified

alternative (Laar and Akça 2007; Lou et al. 2016). Therefore, following Næsset (1997), this study uses accurate tree height measurements obtained from UAV LiDAR, treating tree height itself as the weight to calculate the weighted mean height ($h_w$)



in this study. This method mitigates the limitations of applying Lorey's weighted height without DBH, yielding results that are highly consistent with those obtained using DBH-based ones (Fig. 3). Moreover, the $h_w$ calculation method used in this study is easy to apply to large-scale forest surveys, significantly reducing the input of labor, time, and costs.

Another challenge arises from the regional variations that affect tree height in China, due to the extensive distribution of China's forest. In this study, a variety of variables related to forest growth were considered in a comprehensive way to address this issue (detailed in section 2.3), thereby enhancing the model's explanatory force and improving the accuracy of $h_a$ and $h_w$ estimations. Additionally, forest stand mean height is significantly influenced by tree species (Laar and Akça 2007). However, obtaining accurate tree species information through large-scale and remote sensing-based methods is a challenging (Ørka et

al. 2009). For example, Wu and Shi (2023) considered the distinct nature of forests in different ecological zones in China, as a feasible method in the absence of known tree species, to improve the accuracy of forest canopy height estimation through building ecological zone-based models. Similarly, to avoid the potential boundary effects introduced by multiple specific models, we applied MLME model to capture and differentiate forest types in vegetation divisions across China. It was revealed that 58.1% and 69.3% of the total variances of $h_a$ and $h_w$ were due to variations of vegetation divisions (Table 4). This

indicates significant ecological differences between the vegetation divisions, leading to differences in $h_a$ and $h_w$. Combining the results from Table 5, the MLME method showed a slight performance improvement over ML method in both $h_a$ and $h_w$. Additionally, the improvement in $h_w$ exceeds that in $h_a$, possibly due to its higher ICC as shown in Table 4, which further highlights that $h_w$ is more sensitive to uneven-aged forest stands.

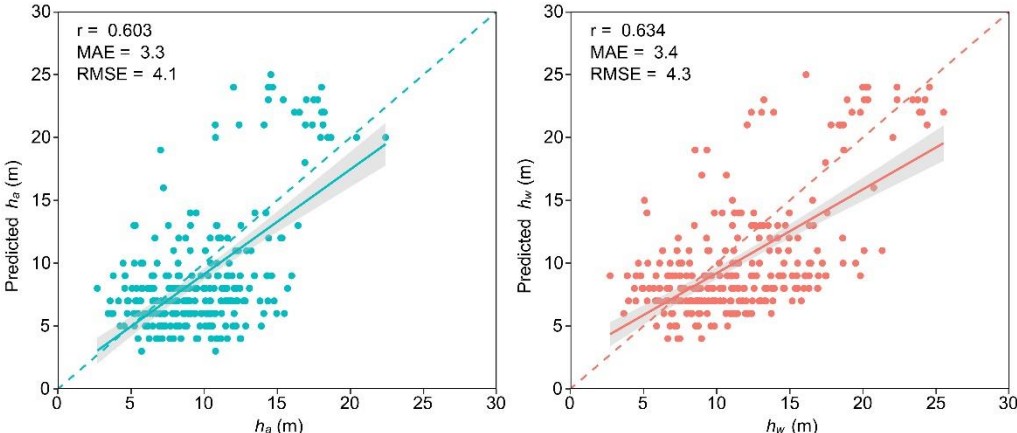

**Figure 12: Accuracy assessment of forest mean heights ($h_a$ and $h_w$) compared with filed measurements.**

The validation results of both $h_a$ and $h_w$ estimations based on a tree-based approach, compared to the field-measured forest stand mean height, show an overestimation for plots with higher tree heights and a slight underestimation for plots with lower tree heights (Fig. 12). The values of r and RMSE for $h_a$ and $h_w$ indicate that $h_a$ and $h_w$, when validated through LiDAR data, exhibit better performance (Figs. 5 and 12). Two reasons are likely to influence the accuracy assessment of $h_a$ and $h_w$ in this

study. First, there may be certain errors in both field-based and LiDAR-based tree height measurements may have contributed.





Variations of researchers and measurement tools, coupled with canopy occlusion can affect the accurate measurements of tree heights, particularly for the tall trees in field (Jurjević et al. 2020; Wang et al. 2019a). In this study, $h_a$ and $h_w$ estimations are influenced by accurate segmentation of individual trees based on LiDAR data. However, the accuracy of LiDAR-based individual tree segmentation is also affected by canopy occlusion (Huo et al. 2022; Li et al. 2012), especially the omission of

low trees obscured by the canopy. This omission of trees with lower heights may increase the influence of the taller trees in $h_w$ weighted by tree height, leading to overestimated results (Laar and Akça 2007; Lefsky 2010)(Fig. 12). Second, the lack of temporal consistency between LiDAR data and field measurements may be another reason. In this study, the LiDAR data were collected from 2015 to 2023, while the field measurements were conducted from 2019 to 2023 (Yang et al. 2023). The relatively long-time span may witness significant tree growth, particularly in young forests (Tang et al. 2014). Therefore, this

temporal inconsistency may have impacted the estimation and validation of $h_a$ and $h_w$. Specifically, it could cause an underestimation of tree heights below 10 m as shown in Figure 12.

Overall, despite the novelty of the data, maps of forest mean stand height, and topic in this study are new, there are still several limitations of the study in terms of close-range LiDAR data and algorithms of LiDAR process. First, as demonstrated in Figure 2, close-range LiDAR data are spatially unevenly distributed across China. While over 1117 km² of close-range LiDAR data

were used, the gaps in data coverage over Qinghai-Tibet Plateau alpine vegetation divisions and uneven distribution in northwestern and southeastern China may influence the mapping accuracy. Enhancing data acquisition and establishing sharing mechanisms for LiDAR data might be key and feasible solutions to address this issue. Second, as shown in Table 1, training and validation data were sourced a nine-year span and collected using various LiDAR sensors. As of 2015, the application of LiDAR has not been widely adopted in forest remote sensing research in China. Considering the cost and the difficulty of data

collection, it was challenging to collect extensive, high-point density and accurate data across China within a short timeframe. Consequently, multiple types of LiDAR sensors were employed over nine years to meet the requirements of data quality. Pre-processing and calibration were conducted to minimize the errors result from variations in LiDAR sensors (Guo et al. 2017; Hu et al. 2021; Zhao et al. 2022). While realizing potential errors associated with tree growth, there is no better alternative data available to achieve this tree-based approach to forest stand mean height mapping. Third, the accuracy of the individual

tree segmentation algorithm based on UAV LiDAR data may be another limitation in this study. Acurate segmentation of individual trees is crucial for forest stand mean height mapping through a tree-based approach. Challenges such as omission and inclusion errors yield throughout individualizing trees in complex stands affect the accurate attribute measurements regarding individual trees, despite visual inspections. The accuracy of individual tree segmentation has also been included in the uncertainty analysis, as shown in Figures. 10 and 11. The development of individual tree segmentation algorithm and

fusion of multi-LiDAR platforms are expected in the future work to enhance individual tree segmentation accuracy.

Earth System
Science
Data

## 5 Data availability

Data described in this manuscript can be accessed at the repository: https://doi.org/10.5281/zenodo.12697784 (Chen et al., 2024).

## 6 Conclusion

We have developed a tree-based approach to create spatially continuous forest stand mean height maps across China through integrating high-point density, high-precision close-range LiDAR data and multisource remote sensing data. The accuracy analysis of $h_a$ and $h_w$ demonstrates the feasibility of the proposed method. A practical framework for forestry investigation based on close-range LiDAR was proposed. The mean values of $h_w$ and $h_a$ are 13.3 ± 3.3 m 11.3 ± 2.9 m on pixel level, respectively. Validation based on LiDAR and field sample data shows that the RMSE values, range from 2.6 to 4.1 m for $h_a$

and 2.9 to 4.3 m for $h_w$, respectively, indicating that our approach outperforms existing forest canopy height maps derived from area-based approaches. Hopefully, our methods and maps will serve as a foundation for estimating carbon storage, monitoring changes in forest structure, managing forest inventory, and assessing wildlife habitat availability.

## Author contributions

Y.C., H.Y. and Q.G. designed the study, Y.C. and H.Y. developed the MLME method, Z.Y., Q.Y., W.L., G.H., Y.R., K.C.,
T.X., M.C., D.L., Z.Q., J.X., Y.Z., and G.X. collected lidar data and provided the filed measurements, all of them processed the lidar data used in this study, Y.C. generated the forest stand mean height products of China and performed the accuracy assessment, H.Y. and Y.C. wrote the original manuscript, and all authors participated in the process of manuscript revisions.

## Competing interests

The authors declare that they have no conflict of interest.

**Disclaimer**

Publisher's note: Copernicus Publications remains neutral with regard to jurisdictional claims made in the text. published maps, institutional affiliations, or any other geographical representation in this paper. While Copernicus Publications makes every effort to include appropriate place names, the final responsibility lies with the authors. Regarding the maps used in this paper, please note that Figs,1, 2, 6, 7, 10 and 11 contain disputed territories.



**Acknowledgements**

We would like to thank all the scientists, engineers, and students who participated in the field observations, instrument maintenance, and data processing. We highly appreciate the valuable and constructive comments on the paper provided by Yanjun Su of Institute of Botany CAS.

**Financial support**

This work was supported by the National Key Research and Development Program [No.2022YFF1300202], and the National Natural Science Foundation of China [grant number 42371329 and 32301285].

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
