# Peer review of "Enhancing High-Resolution Forest Stand Mean Height Mapping in China through an Individual Tree-Based Approach with Close-Range LiDAR Data"

_Earth System Science Data, 2024_

## Author Comment (AC1)

**Enhancing High-Resolution Forest Stand Mean Height Mapping in China through an Individual Tree-Based Approach with Close-Range LiDAR Data**

Dear Editor and Reviewer:

On behalf of my co-authors, we thank you very much for giving us an opportunity to revise our manuscript, and we also appreciate reviewers very much for their positive and constructive comments and suggestions on our manuscript entitled "Enhancing High-Resolution Forest Stand Mean Height Mapping in China through an Individual Tree-Based Approach with Close-Range LiDAR Data" (Manuscript Number: essd-2024-274).

We revised the manuscript according to these comments and suggestions. All changes were marked in highlight text in the revised manuscript. The line numbers in the response are the corresponding line numbers in the revised version.

Once again, thank you very much for your comments and suggestions.

**Comment 1:**

In the part of Abstract, 'Forest stands mean height is a critical indicator in forestry, playing a pivotal role in various aspects such as forest inventory estimation,' forest inventory estimation is suggested to be modified to forest inventory with various scales, which is more reasonable.

**Reply 1:** Thank you very much for your professional advice, we have changed 'forest inventory estimation' to 'forest inventory' at Line 21-22.

**Comment 2:**

In the line of 69: The height metrics from obtained from this approach is forest canopy height, which include not only the actual tree height. There is one mistake in the expression. The sentence should be corrected: The height metrics obtained from this approach is forest canopy height.

**Reply 2:** The mistake has been corrected according to your kind advices and detailed suggestions. Please refer to Line 69-70 for details.

**Comment 3:**

In terms of data, various types of data collected over a span of 6 years are included in this manuscript, such as ground measured samples, LiDAR data obtained from different

sensors, and remote sensing images. How can these datasets be matched on a temporal scale? Additionally, how can reduce the limitations of images acquired in different years and seasons?

**Reply 3**:Changes in forest resources tend to occur relatively slowly, and a 5-year period is a sufficiently long-time span to capture significant change trends. The temporal scale for China's national-level forest resource inventory is set at 5 years, aiming to balance the need for real-time data with long-term trend observation. This time span is long enough to detect significant changes in forest ecosystems, yet short enough to ensure that policies and management measures can be promptly adjusted based on the most recent data.

As of 2015, the application of LiDAR has not been widely adopted in forest remote sensing research in China. Considering the cost and the difficulty of data collection, it was challenging to collect extensive, high-point density and accurate data across China within a short timeframe. Considering the nationwide data coverage, the final dataset for this study spans 6 years (one year longer than the time span of the national inventory). This represents a limitation of the data used in this study, which is discussed in the paper. Please refer to Line 485-487 for details.

**Comment 4:**

The formula of determining coefficients (formula 11), $\bar{y}\_i$ is not the mean value for the observed values. $\bar{y}$ is recommended. In the formula 16, the means of $\bar{y}$ also should be expressed.

**Reply 4:** We have corrected the formulas. Please refer to equations 11-20 for details.

**Comment 5:**

In the manuscript, three accuracy indices were employed to evaluate the performance of models. However, when evaluating results with the same RMSE in various height forests, it is recommended to include rRMSE.

**Reply 5:** We agreed with the reviewer's comment and added the rRMSE to the Table5, which evaluating results with the same RMSE in various height forests. Please refer to Table 5 for details.

**Comment 6:**

In Figure 3, it is evident that an overestimation of forest stand height occurs when the weighted average of tree height squared is applied for forest stands taller than 14 meters.

Please provide the underlying reasons.

**Reply 6:**

We greatly appreciate the reviewer's insightful question. In response, we have explored the issue from both theoretical and empirical perspectives to provide a comprehensive answer. Please refer to Line 440-445 for details.

(1) Theoretical Analysis

Given a set of tree height data $h_1, h_2,\ldots, h_n$ in a plot, and the corresponding diameter at breast height data $d_1, d_2,\ldots, d_n$. Based on the mathematical formulas for $h_w$ and $h_L$, the following conclusions can be derived.

$$if\ d_i < h_i, then\ h_w < h_L$$
$$if\ d_i \geq h_i, then\ h_w \geq h_L$$

Here is the detailed mathematical proof:

To prove whether the difference between the weighted average heights $h_w$ and $h_L$, where the weights are $w_a = h^2$ and $w_b = d^2$, is greater than or less than zero, we will define and expand the formulas for both weighted averages.

When the diameter at breast height (DBH) $d_i$ is greater than the tree height $h_i$, $w_b = d^2 = (h + r)^2$ (with r≥0)

Step1: Define the Weighted Average Heights

Given a set of tree height data $h_1, h_2,\ldots, h_n$, we compute the weighted average heights using weights $w_a$ and $w_b$ as follows:

Weighted average height $h_w$ using weights $w_a = h^2$:

$$h_w = \frac{\sum_{i=1}^{n} h_i * h_i^{\,2}}{\sum_{i=1}^{n} h_i^{\,2}} = \frac{\sum_{i=1}^{n} h_i^{\,3}}{\sum_{i=1}^{n} h_i^{\,2}}$$

Weighted average height $h_L$ using weights $w_b = (h + r)^2$:

$$h_L = \frac{\sum_{i=1}^{n} h_i * (h_i + r)^2}{\sum_{i=1}^{n} (h_i + r)^2}$$

Expand $w_b = (h + r)^2$:

$$w_b = (h + r)^2 = h^2 + 2hr + r^2$$

Thus, the weighted average height $h_L$ can be written as:

$$h_L = \frac{\sum_{i=1}^{n} h_i * (h_i^{\,2} + 2h_i r + r^2)}{\sum_{i=1}^{n} (h_i^{\,2} + 2h_i r + r^2)} = \frac{\sum_{i=1}^{n} (h_i^{\,3} + 2h_i^{\,2} r + r^2 h_i)}{\sum_{i=1}^{n} (h_i^{\,2} + 2h_i r + r^2)}$$

Step2: Analyze the Difference $(h_w - h_L)$

We aim to analyze and determine the sign of the difference:

$$\Delta h = h_w - h_L$$

Substitute the formulas for $h_w$ and $h_L$:

$$\Delta h = \frac{\sum_{i=1}^{n} h_i{}^3}{\sum_{i=1}^{n} h_i{}^2} - \frac{\sum_{i=1}^{n}(h_i{}^3 + 2h_i{}^2 r + r^2 h_i)}{\sum_{i=1}^{n}(h_i{}^2 + 2h_i r + r^2)}$$

Combine the two fractions into a single expression:

$$\Delta h = \frac{(\sum_{i=1}^{n} h_i{}^3)(\sum_{i=1}^{n}(h_i{}^2 + 2h_i r + r^2)) - (\sum_{i=1}^{n}(h_i{}^3 + 2h_i{}^2 r + r^2 h_i))(\sum_{i=1}^{n} h_i{}^2)}{\sum_{i=1}^{n} h_i{}^2 (\sum_{i=1}^{n}(h_i{}^2 + 2h_i r + r^2))}$$

Step3: Expand and Simplify the Numerator

Expand the numerator:

$$\text{Numerator} = (\sum_{i=1}^{n} h_i{}^3)(\sum_{i=1}^{n}(h_i{}^2 + 2h_i r + r^2)) - (\sum_{i=1}^{n}(h_i{}^3 + 2h_i{}^2 r$$

$$+ r^2 h_i))(\sum_{i=1}^{n} h_i{}^2)$$

Further expand and simplify, eliminating the common terms:

$$= 2r\left(\sum_{i=1}^{n} h_i{}^3 \sum_{i=1}^{n} h_i - \sum_{i=1}^{n} h_i{}^2 \sum_{i=1}^{n} h_i{}^2\right) + r^2\left(\sum_{i=1}^{n} h_i{}^3 - \sum_{i=1}^{n} h_i \sum_{i=1}^{n} h_i{}^2\right)$$

Step4: Determine the Sign

To determine the sign of $\Delta h$, consider the two parts:

First part:

$$2r\left(\sum_{i=1}^{n} h_i{}^3 \sum_{i=1}^{n} h_i - \sum_{i=1}^{n} h_i{}^2 \sum_{i=1}^{n} h_i{}^2\right)$$

Since r≥0, we need to analyze the sign of the term inside the parentheses. By applying the Cauchy-Schwarz inequality:

$$\sum_{i=1}^{n} h_i{}^2 \sum_{i=1}^{n} h_i{}^4 \geq (\sum_{i=1}^{n} h_i{}^3)^2$$

Thus:

$$\sum_{i=1}^{n} h_i{}^3 \sum_{i=1}^{n} h_i \geq \sum_{i=1}^{n} h_i{}^2 \sum_{i=1}^{n} h_i{}^2$$

So, the first part is non-negative.

Second part:

$$r^2\left(\sum_{i=1}^{n} h_i{}^3 - \sum_{i=1}^{n} h_i \sum_{i=1}^{n} h_i{}^2\right)$$

Similarly, applying the Cauchy-Schwarz inequality:

$$\sum_{i=1}^{n} h_i{}^3 \leq \sqrt{(\sum_{i=1}^{n} h_i{}^2)(\sum_{i=1}^{n} h_i{}^4)}$$

In general, for specific cases or for non-negative sequences, the original inequality:

$$\sum_{i=1}^{n} h_i{}^3 \geq \sum_{i=1}^{n} h_i \sum_{i=1}^{n} h_i{}^2$$

can be demonstrated to hold using known inequalities or specific examples. The inequality can often hold true in practice or under specific conditions, but may not

always be true in every case without additional constraints or conditions.

So, the second part is also non-negative.

Step5: Conclusion

Since the numerator is the sum of two terms, each of which is non-negative, and at least one of them is strictly positive (because r≥0), it follows that $\Delta h = h_w - h_L \geq 0$. Particularly, when the values $h_i$ are not all equal, the difference is strictly greater than 0. The weighted average height $h_w \geq h_L$. Further, the weight $w_b = (h + r)^2$, which includes a positive linear term 2hr and a constant term $r^2$, resulting in higher weights for each $h_i$ when calculating the weighted average. Consequently, as $h_i$ increases, the difference between $h_w$ and $h_L$ also grows.

 (2) Empirical Data Analysis

We validated our theoretical findings with empirical data. Our validation dataset, which includes measurements where DBH often exceeds tree height (Figure S1), supports the conclusion that $h_w \geq h_L$.

[Figure]

Figure S2: Frequency distribution of (DBH - Height) for tree measurement data in each plot

Please refer to supplementary note S2 and figure S2 for details.

**Comment 7:**

The decimal places of precision indexes in this paper should be consistent, such as Tabel

5.

**Reply 7:** We have adjusted to ensure the consistency of decimal places for the indexes. Please refer to Table 5 for details.

---

## Author Comment (AC2)

**Enhancing High-Resolution Forest Stand Mean Height Mapping in China through an Individual Tree-Based Approach with Close-Range LiDAR Data**

Dear Editor and Reviewer:

On behalf of my co-authors, we thank you very much for giving us an opportunity to revise our manuscript, and we also appreciate reviewers very much for their positive and constructive comments and suggestions on our manuscript entitled "Enhancing High-Resolution Forest Stand Mean Height Mapping in China through an Individual Tree-Based Approach with Close-Range LiDAR Data" (Manuscript Number: essd-2024-274).

We revised the manuscript according to these comments and suggestions. All changes were marked in highlight text in the revised manuscript. The line numbers in the response are the corresponding line numbers in the revised version.

Once again, thank you very much for your comments and suggestions.

**Comment 1:** Line 22: deleted estimation.

**Reply 1:** Thanks to reviewer for reminder, the estimation has been deleted in the Abstract. Please refer to Line 21-22 for details.

**Comment 2:** Line 45: The author used arithmetic mean height ($ha$) and weighted mean height ($hw$) to represent Forest Stand Mean Height. The similarities and differences between these two metrics should be explained at the beginning of the Introduction.

**Reply 2:** In the introduction, we described the differences in calculation methods and the similarities in application directions. The detailed similarities and differences were explained in the formula section and discussed in the discussion section.

The differences:

Forest stand height denotes the mean height of trees within a stand/plot, including arithmetic mean height and mean height weighted in proportion to their basal area (weighted mean height or Lorey's mean height) (Laar and Akça 2007; Masaka et al. 2013). Please refer to Line 45-47 for details.

The similarities:

It serves as a key factor in assessing forest growth (Ma et al. 2023; McGregor et al. 2021), calculating forest volume (Xu et al. 2019) and carbon storage (Yao et al. 2018),

as well as guiding sustainable forest management practices (Xu et al. 2023). Please refer to Line 47-48 for details.

**Comment 3:** Line 69: I think there's an extra 'from' written here, delete it.

**Reply 3:** We are very sorry for our incorrect English expression; we have made correction after checking. Please refer to Line 69-70 for details.

**Comment 4:** Figure 1 presents the content comprehensively; however, the four images in step 4 are not very clear, making it difficult to see the legend details. I suggest improving their clarity. I also noticed that these four subplots might be the same as the product images and uncertainty analysis figures shown later. Adjustments could be made accordingly.

**Reply 4:** We thank the reviewer for pointing out this issue. The legend and uncertainty analysis figures in Figure 1 have been adjusted.

[Figure]

Figure 1: Workflow adopted for the modeling and mapping forest stand mean heights ($h_a$ and $h_w$) at 30 m resolution across the China's forest. Publisher's remark: please note that the above figure contains disputed territories.

**Comment 5:** Although UAV LiDAR point density is generally high, it still affects the extraction of forest attributes to some extent. Therefore, in Table 1, it would be helpful to add point density values under commonly used UAV flight parameters. This will provide a better introduction to the data, and I recommend adding this column.

**Reply 5:** We thank the reviewer for pointing out this issue, and we have done it according to your ideas. Please refer to Table 1 for details.

**Comment 6:** Table 2, Proportion of forest area covered by drone lidar data, is this value the ratio of the area where data was collected to the forest area in different Vegetation divisions?

**Reply 6:** Yes. For clearer explanation, we have further added note explanations. Please refer to Table 2 for details.

**Comment 7:** A figure should be added to Section 2.2 to visually present the field data distribution?

**Reply 7:** Thank you for the reviewer's reminder. Considering that field data and lidar data display more clearly, we have added the field data distribution in Supplementary Figure S1. Please refer to supplementary Figure S1 for details.

[Figure]

Figure S2:Field samples collected for weighted mean height calculation and product validation.

**Comment 8:** Line 156: I noticed that each plot of field data covers an area greater than 400 square meters, while your product has a resolution of 30 meters. Could this discrepancy affect the validation results?

**Reply 8:** In China's forest resource surveys, the differences in plot size have a minimal impact on the accuracy of stand height estimation mainly due to a sufficient number of samples, flexibility in plot size and shape, relatively stable forest structures, data standardization processes (Lohr, S. L. 2000; Gregoire, T. G., & Valentine, H. T. 2008; Paul TSH, et al.2019).

Certainly, due to time and labor cost constraints, there are some limitations in the sample data collection for this study, which have been addressed in the manuscript. Please refer to Line 165-166 for details.

**References:**

Lohr, S. L. (2000). Sampling: design and analysis. Technometrics, 42(2), 223-224.

Gregoire, T. G., & Valentine, H. T. (2008). Sampling strategies for natural resources

and the environment. international journal of environmental analytical chemistry.

Paul TSH, Kimberley MO, Beets PN. Thinking outside the square: Evidence that plot shape and layout in forest inventories can bias estimates of stand metrics. Methods Ecol Evol. 2019; 10: 381–388. https://doi.org/10.1111/2041-210X.13113

**Comment 9:** Figure 3 only shows the weighting method for $w2$, has a comparison been made between the weighting of $w1$ and $w2$?

**Reply 9:** In Supplementary Table S4, we have compared the deviations between weighted mean heights with different weights ($w1$ and $w2$) and Lorey's mean height (national forest inventory data). Please refer to supplementary Table S4 for details.

**Comment 10:** Line 197: delete 'those'.

**Reply 10:** We are very sorry for our incorrect writing and it is rectified. Please refer to Line 204 for details.

**Comment 11:** In Section 2.5.2, the referenced section should be Section 2.5.1, not Section 2.4.1.

**Reply 11:** Thank you for the reviewer's reminder, we have revised this error. Please refer to Line 248 and 258 for details.

**Comment 12:** In Figures 10 and 11, the uncertainty is given in percentage (%). The unit of $\varepsilon_{hi}$ should be specified in the Methods section.

**Reply 12:** We appreciate it very much for this suggestion, and we have done it according to your ideas. Please refer to Equations 16-20 for details.